# OpenReview forum: "When Embedding-Based Defenses Fail: Rethinking Safety in LLM-Based Multi-Agent Systems"
_ICML.cc/2026/Conference — ICML 2026 regular_

### Official Review · Reviewer_D3U1 · 2026-03-08

**Soundness:** 3
**Presentation:** 3
**Significance:** 3
**Originality:** 3
**Overall Recommendation:** 4
**Confidence:** 5

**Summary:**

This paper studies a key weakness of existing embedding-based or outlier-detection defenses in LLM-based multi-agent systems: they rely on the assumption that malicious messages appear as outliers in embedding space, but this assumption breaks under adaptive near-benign attacks. The paper proposes three types of near-benign attacks—Slow Drift, Benign Wrapper, and Chaos Seeding—and provides theoretical explanations for them. To address the collapse of embedding-space separability, the authors propose constructing a confidence signal based on token-level uncertainty (via top-k aggregation of entropy) and then controlling message propagation through two strategies: confidence pruning and confidence down-weighting. Experimental results show that near-benign attacks substantially weaken embedding-based defenses, while the confidence-guided strategies improve robustness.

**Compliance With Llm Reviewing Policy:**

Affirmed.

**Final Justification:**

I maintain Weak Accept (higher confidence). The rebuttal clarifies deployment feasibility (one-time offline prompt selection; one LLM call per round; no access to embeddings/logits/defense internals) and adds useful baseline comparisons and a joint adaptive-attack stress test, which strengthen the empirical story. Some concerns remain about tighter empirical support for the theory and robustness/utility trade-offs under confidence manipulation, but the contribution is solid and relevant.

**Key Questions For Authors:**

Q1. In implementation, how many LLM calls are needed per round for the near-benign attack? What is the candidate generation / filtering pipeline? Does the attack require access to the embedding encoder or the defense threshold?

Q2. For API-based models such as GPT-4o-mini, where logits are unavailable, can the attack still reliably optimize embeddings to remain close to benign ones? What is the cost?

Q3. Confidence is quite sensitive to temperature and sampling. Is the decoding strategy fixed throughout the experiments? If an attacker adjusts generation to reduce entropy, would the defense still remain effective?

Q4. If an attacker jointly optimizes for benign-looking embeddings, high confidence, and maximum target flipping, what would be the failure mode of the proposed defense? Could the authors provide preliminary experiments or at least a discussion?

**Limitations:**

No. While the paper analyzes signal decay and proposes confidence-guided filtering, it does not clearly discuss limitations and negative societal impact. Please add: (1) reliance on token-level probabilities/logits and API constraints; (2) vulnerability to adaptive “high-confidence near-benign” attackers; (3) utility/false-positive trade-offs from pruning/down-weighting; plus a brief dual-use discussion (attack recipes could be misused) and suggested responsible disclosure/mitigations.

**Strengths And Weaknesses:**

Strengths:

1、The attack design is systematic and intuitive. The three near-benign attacks—gradual round-by-round drift, wrapper-based attacks with short malicious payloads, and attacks that enlarge the dispersion of benign messages—effectively reduce embedding separability and are well matched to MAS settings.

2、The derivation of the acceptance region provides a clear explanation of why threshold-based embedding detection inherently leaves room for evasion, which strengthens the paper’s overall persuasiveness.

3、The proposed defense is simple and easy to deploy. The token-level confidence signal requires no additional training, and both pruning and down-weighting are general system-level control mechanisms, making the method practically attractive.

4、The experiments span multiple tasks, models, and topologies, and provide relatively comprehensive coverage.

Weaknesses:

1、The baseline coverage is somewhat narrow. The paper mainly compares against two embedding- or graph-based defenses, G-Safeguard and GUARDIAN. It would be better to include a broader set of baselines, including simpler but stronger alternatives.

2、The attacks are formally defined as an argmax objective (Flip/Disagree), but the implementation details—such as the search strategy, number of model calls per round, whether the setting is white-box, and the computational cost—are not clearly described. The feasibility and cost of near-benign attacks on API-based models (without logits) also need to be clarified.

3、The confidence signal itself may also be vulnerable to manipulation. Token entropy / confidence can be affected by prompting style or decoding strategy, and an attacker may generate content that is highly confident yet incorrect—for example, by jointly optimizing for benign-looking embeddings, high confidence, and target flipping.

4、The Lipschitz-plus-margin analysis that leads to the neighborhood inclusion result is closer to an existence argument. It would be helpful to add empirical evidence, such as estimating boundary thickness or effective $\tau$ or showing the relationship between margin / threshold changes and attack success rate.

---

> ### Author Rebuttal · Authors · 2026-03-31
>
> Thank you for your constructive review. We are pleased that you found the problem important and recognized several key strengths of our work. Our goal in this paper is to provide both a new understanding of why existing embedding-based MAS defenses can fail under near-benign attacks and a simple, effective alternative based on token-level confidence. Below, we respond to your questions and comments in detail.
>
> ## `W1: Baseline scope and additional defense comparisons`
> Thank you for this thoughtful comment. We chose G-Safeguard and GUARDIAN as baselines because they are two representative SOTA embedding-based MAS defenses. We would also like to clarify that MAS defense is still a relatively new research area, and the **available baselines are quite limited**. In fact, **neither G-Safeguard nor GUARDIAN compares against any MAS defense methods**.
>
> We agree, however, that it is important to understand how the proposed attacks interact with other possible defense paradigms. To this end, we have also evaluated our attacks against LLM-as-a-Judge and rule-based filtering baselines, and we will include these results in the revised paper.
>
> |Method|Clean Acc.&uarr;|Under Attack Acc.&uarr;|
> |---|---:|---:|
> |Rule-based|67.0|20.9|
> |LLM-as-a-Judge|70.2|54.0|
> |G-Safeguard|68.0|39.3|
> |GUARDIAN|67.0|32.6|
> |Ours|68.0|52.7|
>
> Table: Results on MMLU with LLaMA-3.1-8B
>
> ## `Q1/W2: Attack cost and attack deployment pipeline`
> Thank you for the question. We clarify that our near-benign attacks do not require iterative optimization or repeated candidate filtering during deployment. In our implementation, we first perform an offline prompt-selection stage, where candidate attack prompts are generated by the LLM following the formulations in Section 5, and we use 50 randomly sampled examples from the dataset to select an effective prompt template. After this selection step, we keep the same attack prompt template fixed across evaluation samples and tasks.
>
> During the actual attack, **the attacker requires only one LLM call per round** to generate its message under the selected prompt template; there is no additional per-round search over candidates. We also clarify that **our attacks do not assume access** to the embedding encoder or any internal model/defense parameters. The attacker only observes the task input and the communicated messages.
>
> ## `Q2/W2: Applicability to API-based models and attack cost`
> Thank you for the question. **Our attack does not require access to model logits**. It only uses the model’s input and output messages, and is designed to keep malicious messages semantically close to benign ones. Therefore, it remains applicable to API-based models such as GPT-4o-mini.
>
> The cost is also low: we select an effective attack prompt template offline using only 50 samples, and then reuse the same template across evaluation samples and tasks. Thus, the main cost is a small one-time prompt-selection step, rather than expensive per-sample or per-round optimization.
>
> ## `Q3/W2: Defense decoding settings`
> We fix the temperature at 0.7 and use the default top-p sampling strategy (p=1) in all experiments across all tasks.
>
> ## `Q3/Q4/W3: Stronger adaptive attacks targeting both embeddings and confidence`
> Thank you for this important comment. A joint optimization attack in MAS is a particularly challenging threat model and, to the best of our knowledge, has not been explicitly studied in prior work. To address this concern, we construct a stronger adaptive attack that uses an adversarial prompt to encourage high-confidence outputs even when the model is uncertain about the answer. We then incorporate our confidence score into the embedding-based defense G-Safeguard and evaluate it under this setting.
>
> |Method|Clean Acc.&uarr;|Joint Adaptive Attack Acc.&uarr;|
> |---|---:|---:|
> |No defense|65.0|41.2|
> |Emb only|68.0|44.6|
> |Emb + Confidence|68.0|50.8|
>
> Table: Results of joint adaptive attack on MMLU with LLaMA-3.1-8B
>
> The results show that, although performance degrades under this stronger attack, our method still outperforms the embedding-only defense. This suggests that the **confidence signal remains informative even against a stronger adaptive attacker** and can still provide meaningful guidance for defense.
>
> ## `W4: Empirical evidence for the theory`
> Thank you for the comment. We agree that adding empirical evidence to estimate the effective boundary would strengthen the paper. In real defense settings, however, directly estimating the exact margin or effective threshold is difficult for methods such as G-Safeguard because of the complexity of the GNN-based models. Therefore, our current experiments instead validate the main practical consequence of the theory: we explicitly construct near-benign attacks whose embeddings remain as close as possible to benign ones, allowing them to pass the learned defense with high probability. The strong attack success rates reported in Table 1 are consistent with this predicted failure mode.

---

> > ### Author Rebuttal · Reviewer_D3U1 · 2026-04-03
> >
> > Thanks for the detailed rebuttal and additional results. The added comparisons (rule-based and LLM-as-a-judge) and the clarification that deployment uses one LLM call per round with only a one-time offline prompt selection substantially address my baseline-scope and feasibility concerns, and the joint adaptive attack experiment is a useful stress test. I still believe the theory would benefit from stronger empirical corroboration and a clearer discussion of confidence manipulability/utility trade-offs, but overall, I maintain my original assessment.

---

> > > ### Author Response · Authors · 2026-04-06
> > >
> > > We sincerely thank the reviewer for the positive assessment and greatly appreciate the recognition of the paper’s broader contributions in the original review, including the systematic design of the near-benign attacks, the simplicity and practical deployability of the proposed confidence-based defense, and the comprehensive experimental coverage across tasks, models, and topologies.

---

### Official Review · Reviewer_4nWP · 2026-03-09

**Soundness:** 2
**Presentation:** 3
**Significance:** 2
**Originality:** 3
**Overall Recommendation:** 3
**Confidence:** 4

**Summary:**

This paper investigates the vulnerability of embedding-based defenses against misinformation attacks in Large Language Model (LLM)-powered Multi-Agent Systems (MAS). The authors propose three adversarial attacks that fundamentally reduce the distribution disparity between malicious and benign embeddings, thereby evading anomaly detection.

**Compliance With Llm Reviewing Policy:**

Affirmed.

**Final Justification:**

My main concern was not addressed (more scenarios needed). I will therefore lower my score.

**Key Questions For Authors:**

These questions correspond to the above weakness.

1.Could the authors elaborate on why they select reasoning-based multiple-choice questions as the setting? To what extent can the proposed findings and methods be generalized to arbitrary MAS tasks, such as collaborative software engineering or creative writing?

2.Could the authors provide further discussion or empirical evidence on how different agent relationships (e.g., cooperative vs. adversarial/debate) and distinct roles (e.g., a "reviewer" agent vs. a "generator" agent) influence the generalizability of the work?

3.The experiments rely exclusively on open-weight models. Could the authors discuss the reasoning behind this choice? Could you further discuss the generalizability of the attack if the MAS (1) communicate through shared vector space, (2) use black-box flagship backbone LLMs?

**Limitations:**

Yes

**Strengths And Weaknesses:**

Strengths:

The objective of bypassing embedding-based detection in MAS is timely and well-defined. The related work is clearly presented. The paper is well-structured and clearly written.

Weaknesses:

My primary concerns regarding this work are the scope and definition of the MAS systems and tasks (Weaknesses 1 & 2).

1.The MAS tasks explored in this work appear limited to reasoning-based multiple-choice questions, which are functionally classification task or constrained generation task. This simplifies the complex scenarios where MAS are frequently deployed regarding multi-view (e.g., debate) and multi-step (e.g., automated task execution). This setting limits the generalizability of the proposed methods.

2.MAS are typically employed to overcome the limitations of single agents through specifically-designed interaction paradigms, i.e., role-playing with specialized experts with corresponding system prompts, tools, ... (such as MetaGPT, CAMEL, etc.). While the paper discusses three communication topologies, it lacks discussion or empirical evidence on how the relationships or functional roles of different agents impact the proposed methods (and baseline methods). For instance, methods effective against a "collaborative" relationship may behave differently in a "debate" or "adversarial" setting where agents are explicitly prompted to critique one another.

3.The experimental scope is limited to open-weight backbone LLMs. However, (1) What if agents utilize high-quality commercial embedding APIs (e.g., OpenAI's text-embedding-3) and communicate through shared vector space? Do these attacks remain effective? (2) Intuitively, agents powered by state-of-the-art flagship models can correct such misinformation through internal reasoning. Evaluating only open-weight models may overestimate the severity of the vulnerability.

---

> ### Author Rebuttal · Authors · 2026-03-31
>
> Thank you for your constructive review. We are pleased that you found the problem timely and recognition that the paper addresses an important and underexplored vulnerability in LLM-based multi-agent systems. Our goal in this paper is to provide both a new understanding of why existing embedding-based MAS defenses can fail under near-benign attacks and a simple, effective alternative based on token-level confidence. Below, we respond to your questions and comments in detail.
>
> ## `Q1/W1: Beyond multiple-choice reasoning`
> Thank you for this helpful comment. We would first like to clarify that **our evaluation is not limited to multiple-choice classification**. In addition to MMLU, we include GSM8K, which involves open-ended multi-step mathematical reasoning and is evaluated by exact match. We also include BBH, which contains challenging compositional reasoning tasks that are difficult for single agents.
>
> We agree, however, that the paper should better clarify this scope. To further address concern about more realistic MAS scenarios, we have also run additional experiments on InjecAgent dataset, which includes more practical tool-integrated settings such as Financial and Physical. We observe the same qualitative trend: near-benign attacks still degrade embedding-based defenses, while our confidence-guided strategy remains more robust. We will include these results and discussion in the revised paper.
>
> |Method|Clean Acc.&uarr;|After Attack Acc.&uarr;|
> |---|---:|---:|
> |G-safeguard|78.3|68.8|
> |Ours|79.1|74.0|
>
> Table: Results on InjecAgent with LLaMA-3.1-8B
>
> ## `Q2/W2: Beyond debate-style communication`
> Thank you for this insightful comment. Our paper studies a communication-level safety question, and therefore abstracts MAS as agents exchanging messages over a communication graph. Debate-style interaction is one important and standard case in this setting, since agents are explicitly prompted to read, critique, and respond to one another. From this perspective, agent relationships and functional roles primarily affect the strength and pathway of message influence, rather than the underlying failure mode itself.
>
> To strengthen this point, **we have run additional experiments** on MMLU under both debate-style MAS and **role-specialized collaborative MAS**. In the role-specialized setting, we assign different prompts to different agents, including Solver, Reviewer, and Verifier roles. We observe the same qualitative conclusion: embedding-based defenses remain vulnerable to near-benign attacks, while confidence-guided filtering is more robust. We will include these results and expand the discussion in the revised paper.
> |Method|Clean Acc.&uarr;|After Attack Acc.&uarr;|
> |---|---:|---:|
> |G-safeguard|69.0|36.2|
> |Ours|68.6|56.8|
>
> Table: Results on MMLU with LLaMA-3.1-8B on role-specialized Setting
>
> ## `Q3/W3: Beyond open-weight and white-box settings`
> Thank you for this important comment. We would first like to clarify that **the paper does not evaluate only open-weight backbones**: Table 1 includes the black-box API model GPT-4o-mini, and our near-benign attacks remain effective in this setting. In addition, when logits are unavailable, our method derives confidence from the returned top-k token probabilities, which are accessible through the API, so the defense remains applicable to black-box models as well.
>
> For the shared-vector-space setting, our current paper focuses on the standard MAS scenario in which agents communicate through natural-language messages. We therefore do not directly evaluate communication through a shared vector space in this work, and view that as an important direction for future study. We will revise the paper to make this scope clearer and to better discuss the extent of generalization. Thank you for bringing this up!

---

> > ### Author Rebuttal · Reviewer_4nWP · 2026-04-03
> >
> > The authors did not address my main concern in the rebuttal: demonstrating the performance of the proposed method in more scenarios: (a) MAS performing automated complex tasks that beyond the capability of one single agent, and (b) flagship commercial LLMs as backbone LLMs that could potentially correct the misinformation. This prevents me from contextualizing the practical value and technical contribution of this work.

---

> > > ### Author Response · Authors · 2026-04-03
> > >
> > > Thank you for the feedback! We would like to clarify the two points you raise.
> > >
> > > For (a), regarding scenarios beyond the capability of a single agent, we would like to clarify that our paper has evaluated this setting on **BBH** (Table 1), which is a benchmark of complex compositional tasks. We consider BBH representative of tasks **where MAS provides benefits over a single agent**. For example, a single LLaMA-3.1-8B agent achieves 61.1% on BBH, whereas a clean 5-agent MAS achieves 67.0%, corresponding to roughly **a 10% relative improvement**. This suggests that BBH captures the type of MAS scenario raised in the comment.
> > >
> > > For (b), regarding commercial LLMs, we have **included experiments on GPT-4o-mini in our paper**, which we consider a strong black-box commercial model. We agree that evaluating stronger models such as GPT-5 would be valuable, but the cost is substantial: a single run of a 5-agent MAS experiment on MMLU would **cost about $450, around 20 times more than GPT-4o-mini**. Due to this expense, we limit our black-box evaluation to GPT-4o-mini, which we believe remains a reasonable representative commercial LLM.
> > >
> > > We appreciate the reviewer’s thoughtful comments. We hope the clarifications above help show that we have considered the proposed method in broader MAS settings in our paper, which in turn provides a better understanding of why existing embedding-based MAS defenses can fail under near-benign attacks and how such failures can be mitigated. We sincerely hope these clarifications help address the reviewer’s concerns.

---

### Official Review · Reviewer_RLzV · 2026-03-09

**Soundness:** 3
**Presentation:** 2
**Significance:** 2
**Originality:** 2
**Overall Recommendation:** 4
**Confidence:** 4

**Summary:**

The paper investigates the vulnerability of current embedding-based defense mechanisms in LLM Multi-Agent Systems (MAS) and proposes three near-benign attack methods that significantly improve the attack success rate. Furthermore, it introduces a defense method based on token-level confidence, which can effectively compensate for the shortcomings of embedding-based defense mechanisms

**Compliance With Llm Reviewing Policy:**

Affirmed.

**Final Justification:**

Thank you for addressing my concerns.

**Key Questions For Authors:**

1. Solving the argmax in Equations (1)-(4) seems computationally expensive. How many samples or iterations are actually needed to generate these "near-benign" messages? It would be helpful to see the attack success rates compared to baselines under the same computational budget.
2. The attacks assume the attacker knows the defender's embedding space. Do these near-benign attacks still work if the attacker and defender use completely different embedding models?
3. Why use top-k entropy specifically? How does it compare to simpler confidence baselines like maximum token probability or logit margins?
4. Since the proposed attacks specifically target embedding-based defenses, how do they perform against other common defenses, such as LLM-as-a-Judge or rule-based filtering?
5.  Modern models naturally produce higher-entropy tokens during Chain-of-Thought reasoning as they explore different logical paths. Does your token entropy defense distinguish between reasoning tokens and final answer tokens? I am concerned this defense might falsely penalize benign agents that are simply doing complex reasoning.

**Limitations:**

yes

**Strengths And Weaknesses:**

### Strengths:

- The paper provides a thorough analysis of the vulnerabilities in embedding-based defenses. The three proposed "near-benign" attacks targeting these defenses make perfect sense from both theoretical and practical perspectives.
- It proposes a corresponding defense method against these attacks, and extensive experiments demonstrate its effectiveness.

### Weaknesses:

- Although the proposed attack and defense theories are highly effective for this specific problem, the paper lacks a discussion on their practical performance in broader scenarios. Specifically, it does not explore the effectiveness of the attacks when facing a combination of **other defense mechanisms**, nor does it evaluate the proposed defense against other types of attacks.
- The generalizability of the attack methods is questionable. The paper does not provide a detailed discussion on how these attacks perform against **different types of embedding methods**, or in scenarios where similarity ranking is conducted via LLM-generated scoring (LLM-as-a-Judge).
- The fundamental principles and **underlying nature** of using token entropy for defense warrant further discussion.

---

> ### Author Rebuttal · Authors · 2026-03-31
>
> Thank you for your constructive review. We are pleased that you found both the motivation and the technical contributions of our paper meaningful. Our goal in this paper is to provide both a new understanding of why existing embedding-based MAS defenses can fail under near-benign attacks and a simple, effective alternative based on token-level confidence. Below, we respond to your questions and comments in detail.
>
> ## `W1: Defense combinations, and attack coverage`
> Thank you for the comment. We would like to clarify two points. First, we compare against two representative SOTA embedding-based MAS defenses, G-Safeguard and GUARDIAN. A direct combination of these methods is not a natural baseline because they rely on different architectures and training objectives. Second, our evaluation is not limited to the three near-benign attacks introduced in this paper. We also evaluate under the standard attack setting commonly used in prior work, including the evaluation regimes adopted for embedding-based defenses such as G-Safeguard and GUARDIAN.
>
> ## `Q1: Attack cost`
> Thank you for the question. In our implementation, solving the argmax in the Eqs does not require expensive per-instance optimization. Instead, we use 50 randomly sampled examples from the dataset to select an effective attack prompt template following the formulation in Section 5. After this one-time selection step, the same prompt template is fixed and reused across evaluation samples and tasks. Therefore, the practical cost is relatively low.
>
> ## `Q2/W2: Attack effectiveness does not depend on the defender’s embedding model`
> Thank you for the comment. We would like to clarify that **our attacks do not assume access** to the defender’s exact embedding model or any internal model/defense parameters. The key idea behind the near-benign attacks is to keep malicious messages semantically close to benign ones, so that they are likely to appear similar under embedding-based defenses more broadly. In our setting, we leverage the standard sentence-transformer all-mpnet-base-v2 to do the embedding, and G-safeguard leverage sentence-transformers/all-MiniLM-L6-v2, GUARDIAN leverage bert-base-uncased, and the results in table 1 show that our attack can break down their defense in the setting that defender use completely different embedding models.
>
> ## `Q3: Why top-k entropy`
> Thank you for the question. Our use of top-k entropy is motivated by [1], which suggests that focusing on the highest-entropy tokens can provide a more informative uncertainty signal than averaging over all tokens. The goal is not to measure overall response uncertainty, but to capture the minority of tokens where the model is making the most consequential reasoning decisions. These tokens provide a more informative confidence signal for defense.
> [1]:Wang, Shenzhi, et al. "Beyond the 80/20 Rule: High-Entropy Minority Tokens Drive Effective Reinforcement Learning for LLM Reasoning." NeurIPS 2025.
>
> ## `Q4/W2: Additional possible defense baselines`
> Thank you for the question. We focus on embedding-based defenses in the paper because they are among the most representative and practical SOTA defenses for MAS. We agree, however, that it is important to understand how the proposed attacks interact with other defense paradigms. We have therefore evaluated our attacks against LLM-as-a-Judge and rule-based filtering baselines, and we will include these results in the revised paper.
>
> |Method|Clean Acc.&uarr;|Under Attack Acc.&uarr;|
> |---|---:|---:|
> |Rule-based Filter|67.0|20.9|
> |LLM-as-a-Judge|70.2|54.0|
> |G-Safeguard|68.0|39.3|
> |GUARDIAN|67.0|32.6|
> |Ours|68.0|52.7|
>
> Table: Results on MMLU with LLaMA-3.1-8B
>
> LLM-as-a-Judge can achieve comparable accuracy but comes with substantially higher token cost (one LLM call pre round per agent), which is a major limitation in multi-round multi-agent settings. Rule-based filtering is often brittle and difficult to generalize across tasks and attack styles.
>
> ## `Q5: Tokens leveraged for defense`
> Thank you for the question. We agree that higher-entropy tokens can naturally arise during CoT reasoning and do not necessarily indicate malicious behavior. In our paper, we do not use the full reasoning trace when computing the confidence score. Instead, we focus on the post-reasoning answer segment—the final answer tokens and the short reasoning span immediately surrounding them.
>
> ## `W3: Motivation for token-entropy defense`
> Thank you for the comment. Embedding-based defenses rely only on the final text representation, which can become unreliable when malicious messages are crafted to remain close to benign ones in embedding space. In contrast, token entropy captures model internal generation uncertainty and provides complementary evidence beyond embeddings. Our goal is not to use token entropy as a standalone maliciousness detector, but as a control signal for message propagation when embedding separability breaks down. We will clarify this motivation in the revision.

---

> > ### Author Rebuttal · Reviewer_RLzV · 2026-04-03
> >
> > Thank you for the detailed rebuttal. The rebuttal partially addresses several concerns. However,  two issues remain unresolved.
> >
> > **Q3:** The rebuttal provides no ablation comparing top-k entropy against simpler confidence baselines such as maximum token probability or logit margin. Given that the defense's effectiveness directly depends on this design choice, the lack of empirical comparison is a meaningful gap. The cited work demonstrates the utility of high-entropy tokens in the context of reinforcement learning for reasoning—but it does not follow that the same signal transfers reliably to adversarial detection
> >
> > **Q5:** The rebuttal claims that the confidence score is computed only over the post-reasoning answer segment, but this is inconsistent with Section 6.1, which defines $U(m)$ over the full token sequence without any segmentation. The rebuttal also does not explain how the "answer segment" is identified in practice—whether by output format tags, a fixed token window, or some other heuristic. This discrepancy remains unresolved.

---

> > > ### Author Response · Authors · 2026-04-03
> > >
> > > We thank the reviewer for the careful reading and thoughtful follow-up comments. We appreciate that the rebuttal addressed part of the original concerns, and below we provide additional empirical evidence and implementation details to directly address Q3 and Q5.
> > >
> > > ### `Q3. Why use top-k entropy?`
> > >
> > > We chose top-k entropy based on the motivation from the *Beyond 80/20* paper, and our empirical results support this choice. To directly address this concern, we conducted an **ablation study on LLaMA-3.1-8B over MMLU and GSM8K**, comparing top-k entropy against simpler confidence baselines.
> > >
> > > In our paper, the default setting uses top-k entropy with \(k = 20\%\), computed over the post-reasoning answer segment, which contains a short answer-focused rationale followed by the final answer token.
> > >
> > > We further evaluate three aspects:
> > >
> > > - **Choice of \(k\)**: 10%, 20%, and 30%
> > > - **Alternative confidence metrics**
> > >   - **Maximum token probability**: uses only the largest probability at each token
> > >   - **Logit margin**: uses only the gap between the top-1 and top-2 logits
> > > - **Different text segments for computing confidence**
> > >   - **Only answer token**: uses only the final answer token (e.g., A/B/C/D or a number)
> > >   - **Top-20 prefix token**: uses only the first 20 tokens of the generated segment
> > >
> > > | Confidence Method | MMLU (%) | GSM8K (%) |
> > > |---|---:|---:|
> > > | None defense | 38.8 | 58.6 |
> > > | Top-k entropy (10%) | 51.9 | **77.9** |
> > > | Top-k entropy (20%) | **52.7** | 77.7 |
> > > | Top-k entropy (30%) | 52.2 | 76.8 |
> > > | Max probability | 49.8 | 74.9 |
> > > | Logit margin | 40.6 | 65.8 |
> > > | Only answer token | 43.7 | 73.6 |
> > > | Top-20 prefix token | 47.9 | 72.8 |
> > >
> > >
> > >
> > > These results show that
> > > * Most confidence signals are indeed helpful, which supports our broader claim that **model-internal uncertainty provides a useful complementary signal for defense**.
> > > * Our **top-k entropy performs best overall**, suggesting that it better captures the localized uncertainty most relevant to reasoning and final answer formation.
> > > * The performance is **not highly sensitive to the exact choice of \(k\)**, which suggests that the method is reasonably stable rather than narrowly tuned.
> > >
> > > ---
> > >
> > > ### `Q5. How is the “message segment” defined in practice?`
> > >
> > > We clarify that, by default, we use the post-reasoning answer segment as the message \(m\) in the equation. This is the **default setting in most MAS papers** because, in thinking/reasoning mode, the reasoning portion is often long and redundant, while the final answer segment typically still contains a short and concise rationale.
> > >
> > > In practice, the answer segment is identified as follows:
> > >
> > > - **Models with an explicit reasoning block** (e.g., **Qwen3**) produce outputs in the form:
> > >   `<think> ... reasoning ... </think> final answer`
> > >
> > >   For these models, we define the answer segment as the content after the special token `</think>`. This final segment typically contains: a short, answer-focused rationale (usually a few sentences), and the final answer token (e.g., A/B/C/D for MMLU).
> > > - **Models without an explicit reasoning block** (e.g., LLaMA and GPT-4o-mini) do not separate reasoning and answer with special tags. For these models, we use the full generated output as the answer segment, since no separate reasoning segment is available.
> > >
> > > Because we restrict the generation length in our experiments, this segment remains short in practice. We will revise the paper to make this implementation detail explicit.
> > >
> > >
> > > We sincerely thank the reviewer again for the constructive feedback. We hope that the additional ablation results and implementation clarifications address the remaining concerns and make our design choices clearer. We respectfully ask the reviewer to reconsider the paper in light of these clarifications.

---

### Official Review · Reviewer_LeAX · 2026-03-13

**Soundness:** 3
**Presentation:** 3
**Significance:** 3
**Originality:** 3
**Overall Recommendation:** 3
**Confidence:** 3

**Summary:**

This paper investigates the safety of LLM multi-agent systems. To overcome the deficiencies in embedding-based defenses, the authors introduce three adaptive attacks, including Slow Drift, Benign Wrapper, and Chaos Seeding. The paper also gives a theoretical result. The paper further argues that defenses should not rely only on text embeddings, and instead proposes two confidence-guided strategies. The proposed confidence-guided method shows better Acc-Avg across all models and datasets compared to G-Safeguard and GUARDIAN.

**Compliance With Llm Reviewing Policy:**

Affirmed.

**Final Justification:**

First, the issue was never whether Lipschitzness is a standard assumption; the issue is whether Theorem 4.1 provides meaningful explanatory value for the specific defenses evaluated in the paper. Second, the authors' clarifications did not adequately distinguish the proposed contributions from prior work in a way that would change my assessment. For these reasons, I maintain my original score.

**Key Questions For Authors:**

1. Theorem 4.1 is stated for an L-Lipschitz scoring function over embeddings. How does this correspond to the actual graph-based detectors being evaluated?
2. Why should token entropy be expected to remain discriminative under adaptive attack? Can an attacker explicitly optimize for both benign-looking embeddings and high confidence, thereby defeating the proposed defense too? The discussion hints at future work, but this seems central to the current method.
3. What does the exclamation mark in (j! →!i) in line 279 mean? Do you mean j $\neq$ i?

**Limitations:**

1. The paper should be more explicit that this is a white-box setting.
2. For a paper on adversarial attacks and defense failures in multi-agent LLM systems, saying there are no societal consequences that need highlighting is not adequate.

**Strengths And Weaknesses:**

Strengths:

1. The paper targets an underexplored problem in MAS safety evaluation.
2. The paper is easy to follow and is supported by both theory and experiments.
3. The experiments are suitable for the claims.

Weaknesses:

1. The theorem depends on a Lipschitz constant and a benign margin, and then concludes that the acceptance region contains a neighborhood around the benign support. This is mathematically sound, but it does not characterize the actual deployed defenses. It does not really model the full graph-based detector, temporal dynamics, or training procedure of G-Safeguard / GUARDIAN. As a result, the theory motivates the paper, but does not establish the failure of the real methods.
2. Confidence pruning and confidence down-weighting are sensible, but they are also heuristics rather than a new defense principle. The main novelty lies in the evaluation setting and the diagnosis of embedding failure, not in the defense itself. That is acceptable, but the paper should not overstate methodological novelty.
3. The paper treats token-level entropy as a useful signal, but does not analyze when confidence is actually trustworthy under adversarial prompting.

---

> ### Author Rebuttal · Authors · 2026-03-31
>
> Thank you for your constructive review. We are pleased that you found the paper clear, well supported by both theory and experiments. Our goal in this paper is to provide both a new understanding of why existing embedding-based MAS defenses can fail under near-benign attacks and a simple, effective alternative based on token-level confidence. Below, we respond to your questions and comments in detail.
>
> ## `Q1/W1: Relation between Theorem 4.1 and practical graph-based detectors`
> Thank you for the comment. Theorem 4.1 is a general formulation and does not attempt to capture every architectural or optimization detail of a graph-based detector. Our goal is instead to **abstract the common pipeline shared by embedding-based defenses**: messages are first mapped to embeddings and ultimately produce a decision score (malicious vs. benign). Therefore, we use Lipschitzness as a smoothness abstraction for this pipeline to motivate our study.
>
> We would like to emphasize that normalized graph propagation is continuous and Lipschitz, and that even if some graph-based detectors are not globally Lipschitz, they are **typically locally Lipschitz**. This makes our theory a natural smooth approximation for many practical graph-based neural detectors, including detectors such as G-Safeguard and GUARDIAN. In this sense, our theory is intended to correspond to such practical graph-based detectors and to explain the failure mode of these methods under near-benign attacks.
>
> We will include this clarification in the revised paper and make it more explicit. Thank you!
>
> ## `Q2/W2: Clarifying the contribution of the confidence-guided defense`
> Thank you for the comment. We would like to clarify the contribution of this paper. We identify and characterize the near-benign failure mode of existing MAS defenses, a limitation that, to our knowledge, has not been explicitly recognized or studied in prior work. Building on this finding, we further show that token-level confidence provides a complementary signal precisely when embedding-space separability breaks down. This connection is non-obvious and, to our knowledge, has not been explored in the MAS safety literature. The proposed confidence-guided pruning/down-weighting methods are therefore presented as simple and practical instantiations of this broader insight: when embedding-space signals become unreliable under near-benign attacks, model-internal signals can provide complementary evidence.
>
> ## `W3: Stronger attack targeting on both embedding and confidence`
> Thank you for the comment. We agree that whether our confidence-guided defense remains informative under adaptive attack is a central question. A joint optimization attack in MAS is a particularly challenging threat model and, to the best of our knowledge, has not been explicitly studied in prior work. To address this concern, we construct a stronger adaptive attack that uses an adversarial prompt to explicitly encourages the model to produce high-confidence outputs even when its answer is unreliable. We then incorporate our confidence score into the embedding-based defense G-Safeguard and evaluate it on MMLU under this setting.
>
> |Method|Clean Acc.&uarr;|Joint Adaptive Attack Acc.&uarr;|
> |---|---:|---:|
> |No defense|65.0|41.2|
> |Emb only|68.0|44.6|
> |Emb + Confidence|68.0|50.8|
>
> Table: Results of joint adaptive attack on MMLU with with LLaMA-3.1-8B
>
> The results show that, although performance degrades under this stronger attack, our method still outperforms the embedding-only defense. This suggests that **the confidence signal remains informative even against a stronger adaptive attacker** and can still provide meaningful guidance for defense. We will revise the paper to include this experiment and make this limitation explicit.
>
>
> ## `Q3: Notation in line 279`
> Thanks for pointing this out. We apologize for this LaTeX formatting typo, and we will correct it in the revised paper.
>
> ## `L1: Clarifying the scope beyond white-box settings`
> Thank you for the comment. We would like to clarify that our method is **not restricted to a white-box setting**. In addition to open-weight models, we also report results on the black-box API model GPT-4o-mini in Table 1, demonstrating the effectiveness of both our attack and defense methods in this setting. For black-box models, as stated in the paper, we derive confidence from the returned top-k token probabilities, obtained through the OpenAI API, when logits are unavailable.
>
> ## `L2: Societal impact`
> Thank you for the comment. We would discuss the potential societal impacts of our work, and will add this paragraph in the revised paper:
> >Since the paper introduces stronger attack strategies, it also carries a degree of dual-use risk: these ideas could potentially be misused to probe or evade weak defenses. However, our goal is to expose realistic vulnerabilities that are important for robust evaluation and improved defense design. We hope this work will motivate stronger defenses for future MAS.

---

> > ### Author Rebuttal · Reviewer_LeAX · 2026-04-03
> >
> > The rebuttal addresses several points, notably by adding an adaptive attack experiment. However, some of my concerns remain. The connection between theory and practice is too weak, and the methodological novelty is limited. As a result, I will maintain my original assessment.

---

> > > ### Author Response · Authors · 2026-04-06
> > >
> > > Thank you for the feedback. We appreciate that the rebuttal addressed several of the concerns. We would like to clarify two points regarding the theory and the methodological novelty.
> > >
> > > ### `Regarding the Theory`
> > >
> > > Thank you for the feedback. We clarify that our theory serves as a natural smooth approximation of the common scoring pipeline used by many practical graph-based neural detectors, including methods such as G-Safeguard and GUARDIAN. **Such smooth abstractions are standard in modern ML theory**: prior work studies message-passing GNNs through Lipschitz continuity and stability [1,2,3], and more broadly analyzes practical neural networks through local Lipschitz regularity, as in classical works such as Wasserstein GAN and Neural ODE [4,5]. Therefore, our use of Lipschitzness is not ad hoc; rather, it follows a common theoretical strategy of using smoothness to capture the essential behavior of practical neural architectures while abstracting away lower-level implementation details.
> > >
> > >
> > > >[1] Gama et al., Stability Properties of Graph Neural Networks, IEEE Transactions on Signal Processing 2020.
> > > >
> > > >[2] Verma et al., Stability and Generalization of Graph Convolutional Neural Networks, KDD 2019.
> > > >
> > > >[3] Levie et al., A graphon-signal analysis of graph neural networks, NeurIPS 2023.
> > > >
> > > >[4] Arjovsky et al., Wasserstein Generative Adversarial Networks, ICML 2017.
> > > >
> > > >[5] Chen et al., Neural Ordinary Differential Equations, NeurIPS 2018.
> > >
> > >
> > >
> > > ### `Regarding the Novelty`
> > > Thank you for the feedback. We emphasize that the core methodological novelty of our defense approach is the **discovery of a new principle** for MAS safety: **model-internal signals provide an effective complement to exisiting defenses**. This principle is especially important because prior MAS defense work has relied exclusively on external message embeddings, and such methods break down under near-benign attacks.
> > >
> > > We validate this principle through our lightweight, training-free defense approach. Although this concrete design is itself a contribution, the larger contribution is the defense principle it establishes and validates. This perspective **opens up a new and previously unexplored direction for MAS defense**. More fundamentally, the paper redefines the design space of MAS defense: robust defense cannot rely on embeddings alone and must incorporate model-internal signals as a core component of the defense pipeline.
> > >
> > > Beyond the defense-side novelty, we **appreciate that the reviewer recognized the novelty of our proposed near-benign attacks**, which directly reveal when and why embedding-based defenses fail. As noted in the original review, “The main novelty lies in the evaluation setting and the diagnosis of embedding failure.”
> > >
> > > We sincerely thank the reviewer for the feedback and would greatly appreciate reconsideration of our paper in light of these clarifications.

---

### Decision · Program_Chairs · 2026-04-30

**Decision:**

Accept (regular)

**Comment:**

This paper exposes vulnerabilities in embedding-based defenses for LLM multi-agent systems (MAS), introducing "near-benign" attacks and a practical token-level confidence defense.

The reviewers provided mixed scores (4,4,3,3; two Weak Accepts, two Weak Rejects). Supporters (RLzV, D3U1) praised the systematic attack design and the defense's simplicity. Detractors (LeAX, 4nWP) raised valid concerns regarding methodological novelty, the gap between the theoretical abstractions and deployed graph-based detectors, and the limited scope of the evaluated MAS scenarios.

Despite the split scores, I recommend a Weak Accept. The authors' rebuttal successfully addressed the most critical empirical concerns. They provided crucial ablations on top-k entropy, stress-tested the defense against joint adaptive attacks, and expanded evaluations to role-specialized and tool-integrated settings. While the theoretical formulation of Lipschitz smoothness does not perfectly capture all practical defense complexities, identifying the near-benign failure mode is a highly actionable and timely contribution to the MAS safety community.